# Testing the cultural-invariance hypothesis: A global analysis of the relationship between scientific knowledge and attitudes to science

**Patrick Sturgis**[1]*, **Ian Brunton-Smith**[2], **Nick Allum**[3], **Simon Fuglsang**[4]

**1** Department of Methodology, London School of Economics and Political Science, London, United Kingdom, **2** Department of Sociology, University of Surrey, Guildford, Surrey, United Kingdom, **3** Department of Sociology, University of Essex, Wivenhoe Park, Essex, United Kingdom, **4** Department of Political Science, Aarhus University, Aarhus, Denmark

☯ These authors contributed equally to this work.
\* p.sturgis@lse.ac.uk

## Abstract

A substantial body of research has demonstrated that science knowledge is correlated with attitudes towards science, with most studies finding a positive relationship between the two constructs; people who are more knowledgeable about science tend to be more positive about it. However, this evidence base has been almost exclusively confined to high and middle-income democracies, with poorer and less developed nations excluded from consideration. In this study, we conduct the first global investigation of the science knowledge-attitude relationship, using the 2018 Wellcome Global Monitor survey. Our results show a positive knowledge-attitude correlation in all but one of the 144 countries investigated. This robust cross-national relationship is consistent across both science literacy and self-assessed measures of science knowledge.

## Introduction

The association between scientific knowledge and attitudes to science and technology has been the subject of heated debate since the inception of the study of public understanding of science. The simple claim at the centre of this debate is that higher levels of science knowledge produce more positive evaluations of science. This has come to be known, somewhat pejoratively, as the 'deficit model' of public attitudes to science [1, 2]. From this perspective, public resistance to controversial areas of science and technology arises from a deficiency in scientific awareness and understanding. Where science knowledge is low or non-existent, the argument goes, fear of the unknown and mistaken beliefs drive negative responses to scientific research and technology.

In a landmark study in this area, Allum et al. (2008) investigated the strength and generality of the knowledge-attitude relationship using a meta-analysis of existing surveys [3]. They identified a small positive correlation, the magnitude of which varied according to the focus and specificity of the attitudinal measure considered. However, as with most attitudinal studies, the

**Data Availability Statement:** All relevant data and codes for all analyses are deposited at the corresponding author's Open Science Framework

page (DOI 10.17605/OSF.IO/ARGEP). The data are also publicly available at the Wellcome Trust website: https://wellcome.org/reports/wellcome-global-monitor/2018

**Funding:** The author(s) received no specific funding for this work.

**Competing interests:** The authors have declared that no competing interests exist

range of countries included was limited to high- and middle-income democracies, covering only a minority of peoples and cultures across the world [4]. The narrow range and homogeneity of the countries available in the 2008 data led these authors to call for a broader investigation of what they termed the 'cultural invariance hypothesis'; the claim of a (near) universal positive relationship between science knowledge and attitudes towards science across countries, socio-political contexts, and cultures.

That task is the object of this study: to revisit the science knowledge-attitude nexus, but in a data set with far higher coverage of countries and societal contexts across the world. To do this, we use the 2018 Wellcome Global Monitor (WGM) survey, which includes over 149,000 respondents across 144 countries. The content of the WGM questionnaire enables us to estimate this correlation for two different measures of science knowledge commonly used in the existing literature: one which taps 'science literacy' [1] and another which uses respondents' self-assessments of their understanding of science [5].

## Background and relevant literature

When scholars in the interdisciplinary field of public understanding of science began to assess the knowledge-attitude relationship some 40 years ago [6–8], they launched a strand of theoretical and empirical research on the nature and role of scientific knowledge in the formation of attitudes towards science and technology. Why was this focus on science knowledge thought to be interesting and worthwhile? Arguably it is because it embodied the default assumption of many policy makers and scientists about the roots of public support for (and opposition t0) scientific research programmes [9]; that 'to know science is to love it' [3]. And, by the same token, that public resistance to controversial areas of science is grounded in ignorance and misunderstanding, an orientation which remains prominent to this day [10, 11].

The basic premise of the deficit model's proposition has been supported by findings from surveys in the US and Europe which showed that large majorities of the public were unable to recognise basic scientific facts, such as that the earth orbits the sun, or that electrons are smaller than atoms [12, 13]. Allum et al's 2008 meta-analysis [3] demonstrated firstly that, on average, a small positive correlation existed between science knowledge and attitudes across high and middle-income countries. Secondly, it showed that the magnitude, and even the sign and of the correlation, varied according to the specific focus of the science–with some areas exhibiting considerably stronger correlations than others. The significance of these findings could be interpreted in two ways. On the one hand, they provided support to those who contend that science literacy matters for promoting public enthusiasm for science and technology. On the other hand, the very same findings offered evidence that science knowledge explains only a small amount of the total variation in science attitudes. This left space for other more important drivers of science attitudes, such as identity [14–16], transparency [17–19], and societal shocks [20, 21], to be foregrounded.

There have been a number of further studies since 2008 that have examined the knowledge-attitude correlation across a range of scientific and technological contexts. These have mostly continued to find positive correlations. Studies have found, for example, that science knowledge increases support for research on stem cells [22], climate change [23, 24], evolution and nanotechnology [25], as well as general attitudes to science [26]. A smaller number have found negative correlations, usually when considering specific population sub-groups and for areas of science that are subject to political controversy or ethical debate [27, 28]. For instance, Cacciatore et al [29] found that perceptions of risk about biofuels were higher amongst those who knew more about the science underpinning biofuel extraction. A negative or null relationship has also been found when issue-relevant predispositions are included as moderators of the

knowledge attitude relationship [14, 25, 30, 31]. In these cases, while a positive effect of knowledge is observed for the population as a whole, it is zero or negative for groups such as conservatives and those with religious convictions.

While the moderating effect of ideology and values on the knowledge-attitude nexus is now well-established, the influence of country or other geospatial units has been afforded less attention in the empirical literature. An exception is Bauer et al [32] who found a curvilinear relationship between the strength of the knowledge–attitude correlation at the country level with gross domestic product (GDP). The correlation they observed, was lowest in European countries that are most economically advanced but markedly stronger in less developed societies. Bauer and colleagues' interpretation of this relationship was that citizens of countries at an earlier stage of development are in thrall to the power of science and its potential for social and economic transformation. In 'post-industrial' nations, however, science and technology is taken for granted as a commonplace feature of everyday life and the public begins to question the ethics of research programmes and the often uneven distribution of their benefits [33]. Allum et al [34] confirmed this finding with a similar study showing considerable variation in the knowledge-attitude correlation. They found, additionally, that the heterogeneity they observed was partially explained by indicators of regional economic and technological development.

Our objective in this paper is to return to the question of how scientific knowledge is related to attitudes to science but now adopting a truly global perspective. We fit multi-level models to survey data covering over 90% of the world's population, providing by far the most comprehensive coverage of global attitudes to date. We assess the overall strength and direction of the knowledge-attitude relationship, as well as how it varies across country contexts and using two different measures of science knowledge.

## Data and measures

We use data from the 2018 Wellcome Global Monitor, a cross-national survey of adults aged 15+ living in households at non-institutional addresses. The achieved sample size was approximately 1,000 in each of the 144 countries, rising to 2,000 for China, India, and Russia, resulting in a total sample of 149,014 individuals. In countries with at least 80% phone coverage, interviews were carried out via Computer Assisted Telephone Interviewing (CATI), with face-to-face interviewing used in the remaining countries. For telephone interviews, sampling was implemented through either Random Digit Dialling (RDD) or simple random sampling from nationally representative lists of numbers. Dual frame sampling was used in countries with high rates of mobile phone penetration. Sampling for in-home interviews was implemented in 2-stages, where the first stage selected primary sampling units (PSU) with probability proportional to population size and the second stage selected a random sample of households within each PSU, using the random route method. The source questionnaire was produced in English, Spanish, and French and then translated using local translators into every language spoken by more than 5% of the resident population in each country using back translation. Further details about the methodology of the GWP can be found in the survey technical report (https://cms.wellcome.org/sites/default/files/wgm2018-methodology.pdf).

We include two measures of science knowledge, one that approximates the standard type of science literacy index used in most existing studies, and a measure of self-assessed understanding of science. While covering the same underlying conceptual domain, there are important differences between these types of measures [35, 36]. Our purpose in including both is not to assess their relative performance in terms of validity and reliability but rather to provide a full

descriptive picture of the global science knowledge-attitude association from the perspective of existing research on this question.

The science literacy measure is derived from three items which tap, directly or indirectly, the respondent's understanding of scientific concepts. It is taken as the predicted score from a 2-parameter Item Response Theory (IRT) model from the following three items (correct answers indicated (1) incorrect (0)):

- *Do you think studying diseases is a part of science*? Yes (1), No (0)

- *On this survey, when I say 'science' I mean the understanding we have about the world from observation and testing. When I say 'scientists' I mean people who study the Planet Earth, nature and medicine, among other things. How much did you understand the meaning of 'science' and 'scientists' that was just read*? A lot (1), Some (0), not much (0), not at all (0)

- *A vaccine is given to people to strengthen their body's ability to fight certain diseases. Sometimes people are given a vaccine as [insert country equivalent term for a shot or an injection], but vaccines can also be given by mouth or some other way. Before today, had you ever heard of a vaccine*? Yes (1), No (0)

We considered a fourth item which asked 'do you think poetry is a part of science? (Yes(1), No(0)) but the IRT model indicated that this did not scale with the other three items. These items were not designed with the intention of measuring science literacy and the scale is suboptimal in both content coverage and specificity. We return to a consideration of the implications of these limitations in the discussion section.

Self-assessed science knowledge is measured with a single item, '*How much do you*, *personally*, *know about science*? *Do you know a lot*, *some*, *not much*, *or nothing at all*'.

The measure of general attitude to science was also derived using a 2-parameter IRT model applied to the following three items:

- *In general, do you think the work that scientists do benefits most, some, or very few people in this country*? A lot (1), Some (0), Not much (0), Not at all (0).

- *In general, do you think the work that scientists do benefits people like you in this country*? Yes (1), No (0)

- *Overall, do you think that science and technology will help improve life for the next generation*? Yes (1), No (0)

Full details of the IRT models for the science literacy and attitude to science measures can be found in S1-S3 Tables and S1-S3 Figs in the S1 File and histograms of the two knowledge and the attitude to science measures for all countries in S1-S6 Figs in the S1 File. Data and code for all analyses are deposited at the corresponding author's Open Science Framework page (https://osf.io/argep/).

## Analysis strategy

We estimate multilevel (hierarchical linear) models of the following general form:

$$y_{ij} = \beta_0 + \beta_1 x_{ij} + x'_{ij}\beta + z'_j\beta + u_{0j} + u_{1j} + e_{ij} \qquad (1)$$

Where $y_{ij}$ is science attitude and $x_{ij}$ is a measure of scientific knowledge for individual $i$ in country $j$, and $x'_{ij}\beta$ and $z'_j\beta$ are the vectors of individual and country-level covariates, respectively. The intercept and the coefficient for science knowledge are allowed to vary between countries, with random effects $u_{0j}$ and $u_{1j}$. The variances of the random effects, $\sigma^2_{u0}$ and $\sigma^2_{u1}$ are

assumed to be bivariate normally distributed with covariance $\sigma_{u01}$. The individual level covariates are sex, age, income (in quintiles), employment status, and whether or not the respondent had learned about science at primary school, secondary school and/or college/university. At the country-level we adjust for measures of population size, GDP per capita, income inequality (the GINI coefficient) and education level (the Harmonised Learning Outcome scale). These are taken from the world bank indicators database(https://databank.worldbank.org/home.aspx). Models are fitted using the lme4 package in R version 4.3.1. Data and R code for replication are available at the corresponding author's Open Science Framework page (https://osf.io/argep/).

## Results

The global distributions for science literacy and self-assessed knowledge across countries are shown as maps in Fig 1 (country means for both knowledge measures are reported in S4 Table in the S1 File. For science literacy, scores are highest in Hungary (69%), USA (67%), Canada (63%) and lowest in Vietnam (1%), Cambodia (1%) and Pakistan (1%). Overall, between country differences account for 20% of the total variation in this science knowledge measure across countries (estimated from an 'empty' multilevel model).

For self-assessed knowledge we show the percentage reporting they know 'a lot' about science, with the US again ranking at the top end, with 22% of Americans selecting this response alternative, followed by Denmark (21%) and Lebanon (17%). Self-assessed knowledge is lowest in Myanmar (1%), Vietnam (1%) and Tanzania (1%).

The US ranking at the top of the science knowledge distribution is consistent with previous research [37] and may be attributed to the greater focus on a broad-based science curriculum until later in a US high school education compared to school systems elsewhere [38]. At 14%, between country differences account for somewhat less of the total variability in self-assessed science knowledge, suggesting that this measure is more dependent on individual than societal level influences. Fig 1 confirms that both measures of science knowledge vary quite markedly across countries, with some countries scoring very low and others scoring toward the maximum of the distribution. It is notable that more economically developed societies appear to have somewhat higher scores than middle- and lower-income countries. Next, we turn to a consideration of whether this global heterogeneity in science knowledge is reflected in cross-country differences in the knowledge-attitude correlation.

The results from the multi-level models are presented in Table 1. Model 1 is an 'empty' model which enables us to estimate the unconditional between-country variance component of 10% (0.062/0.533+0.062). This is somewhat lower than the between country variance of the two science knowledge measures but nonetheless indicates there is sufficient variability to warrant proceeding to explanatory models. Model 2 estimates the unconditional association between science literacy and general attitude to science at 0.244 (p<0.001). Controlling for the full set of individual and country level covariates (model 3) reduces this coefficient slightly, with the science literacy coefficient now estimated as 0.194 (p<0.001). Standardising the science literacy estimate from model 3 produces a coefficient of 0.158, which is remarkably close to the equivalent estimate of 0.14 in the 2008 Allum et al meta-analysis [3], albeit covering a much larger and more diverse range of country contexts.

Models 4 and 5 include the equivalent parameter estimates for self-assessed science knowledge. The pattern here is similar, with a significant positive coefficient across both model specifications. Introducing the covariates has a larger impact on the self-assessed knowledge coefficient, reducing it by 30% from 0.139 to 0.096. Again, this suggests that respondent self-assessments of their scientific understanding are more a function of these characteristics than is the science literacy measure.

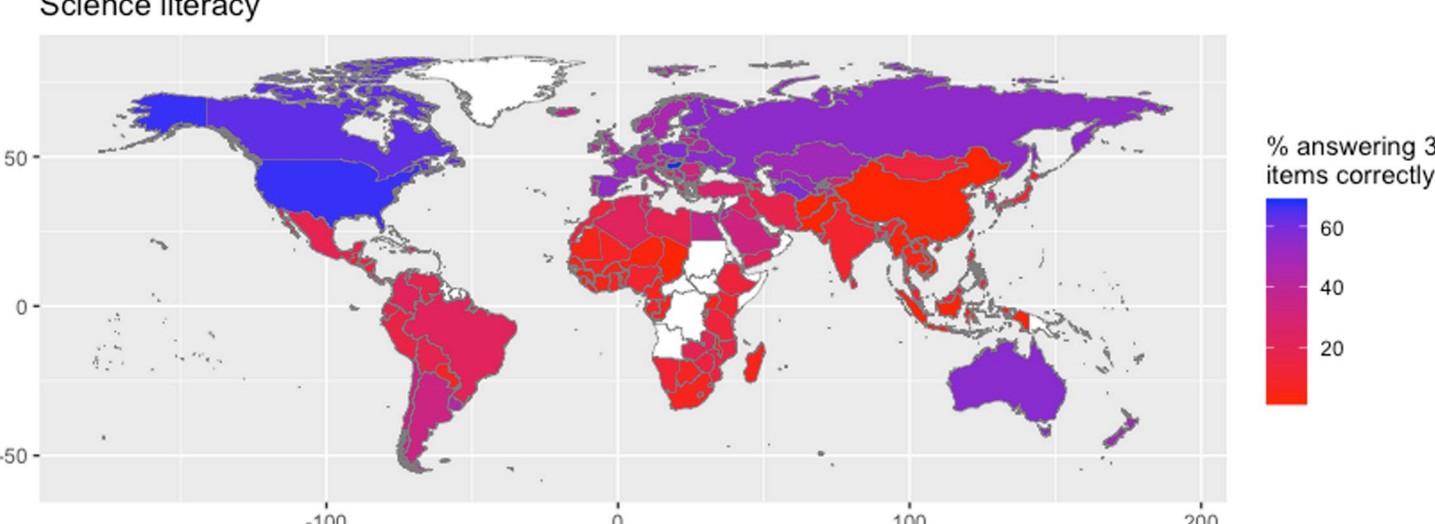

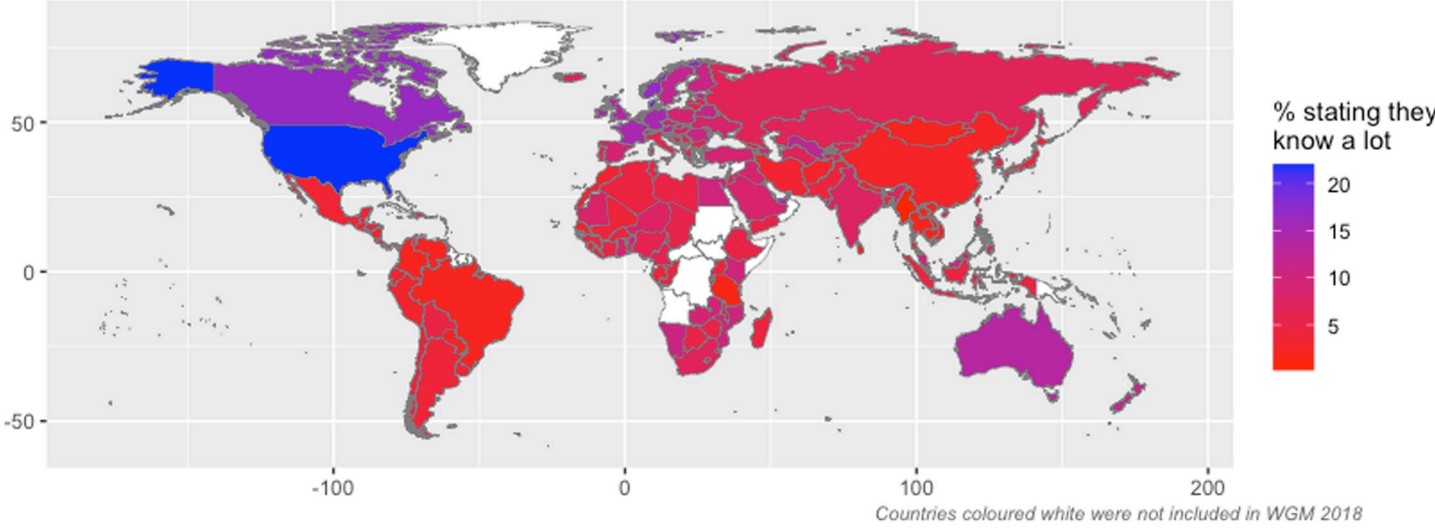

**Fig 1. Global distribution of science knowledge.**

Model 6 shows that science literacy and self-assessed knowledge are independently related to general attitude to science, supporting the conclusion of Rose et al [5] that they represent distinct, though partially overlapping, conceptual domains. When including both science knowledge measures in the same model, we see that the coefficient for science literacy is essentially unchanged, while the coefficient for self-assessed knowledge is reduced by more than 40%. This suggests that a large part of the self-assessed knowledge measure is determined by an individual's actual level of science knowledge. How people assess their level of knowledge, on the other hand, makes no discernible contribution to how much they know about science, a pattern which makes intuitive sense.

The significant random coefficients in models 3 and 5 (0.01 for science literacy, 0.003 for self-assessed knowledge) indicate that the knowledge-attitude correlation differs across countries. Fig 2 plots the estimated difference in general attitude to science between those with the

**Table 1. Multilevel regression models predicting general attitude to science.**

| | Model 1 | Model 2 | Model 3 | Model 4 | Model 5 | Model 6 |
|---|---|---|---|---|---|---|
| Science literacy | | 0.244 *** | 0.194 *** | | | 0.174 *** |
| | | (0.010) | (0.010) | | | (0.010) |
| Self-assessed knowledge | | | | 0.139 *** | 0.096 *** | 0.070 *** |
| | | | | (0.006) | (0.006) | (0.005) |
| Male | | | 0.029 *** | | 0.017 *** | 0.019 *** |
| | | | (0.004) | | (0.005) | (0.005) |
| Age (10yrs) | | | -0.001 | | -0.002 | -0.000 |
| | | | (0.001) | | (0.001) | (0.001) |
| Secondary education | | | -0.042 *** | | -0.044 *** | -0.047 *** |
| | | | (0.007) | | (0.007) | (0.007) |
| University education | | | 0.008 | | 0.010 | -0.011 |
| | | | (0.009) | | (0.010) | (0.010) |
| Income (Q2) | | | 0.029 *** | | 0.034 *** | 0.030 *** |
| | | | (0.008) | | (0.008) | (0.008) |
| Income (Q3) | | | 0.041 *** | | 0.046 *** | 0.038 *** |
| | | | (0.007) | | (0.008) | (0.008) |
| Income (Q4) | | | 0.055 *** | | 0.061 *** | 0.053 *** |
| | | | (0.007) | | (0.008) | (0.007) |
| Income (Q5) | | | 0.056 *** | | 0.064 *** | 0.052 *** |
| | | | (0.007) | | (0.007) | (0.007) |
| Unemployed | | | -0.056 *** | | -0.059 *** | -0.056 *** |
| | | | (0.009) | | (0.009) | (0.009) |
| Out of the workforce | | | 0.016 ** | | 0.009 | 0.015 ** |
| | | | (0.005) | | (0.005) | (0.005) |
| Science education (primary) | | | 0.074 *** | | 0.083 *** | 0.063 *** |
| | | | (0.006) | | (0.006) | (0.006) |
| Science education (secondary) | | | 0.079 *** | | 0.088 *** | 0.063 *** |
| | | | (0.007) | | (0.007) | (0.007) |
| Science education (college) | | | 0.061 *** | | 0.058 *** | 0.042 *** |
| | | | (0.006) | | (0.006) | (0.006) |
| Population total (std) | | | 0.100 ** | | 0.083 ** | 0.091 ** |
| | | | (0.033) | | (0.032) | (0.032) |
| GDP per capita (std) | | | 0.087 ** | | 0.082 ** | 0.088 *** |
| | | | (0.026) | | (0.026) | (0.026) |
| Harmonised Learning Outcome (std) | | | -0.028 | | 0.008 | -0.015 |
| | | | (0.030) | | (0.029) | (0.029) |
| Gini coefficient (std) | | | -0.022 | | -0.030 | -0.020 |
| | | | (0.023) | | (0.022) | (0.022) |
| (Intercept) | -0.009 | -0.016 | -0.007 | -0.016 | -0.010 | -0.014 |
| | (0.021) | (0.020) | (0.021) | (0.020) | (0.021) | (0.021) |
| **Random effects** | | | | | | |
| Country variance | 0.062 | 0.058 | 0.051 | 0.057 | 0.050 | 0.052 |
| Individual variance | 0.533 | 0.510 | 0.501 | 0.517 | 0.509 | 0.498 |
| Science literacy | | 0.011 | 0.010 | | | 0.010 |
| Covariance (with Country variance) | | -0.002 | -0.001 | | | -0.000 |
| Self-assessed knowledge | | | | 0.005 | 0.003 | 0.002 |
| Covariance (with Country variance) | | | | -0.001 | -0.002 | -0.003 |

*(Continued)*

**Table 1.** (Continued)

|  | Model 1 | Model 2 | Model 3 | Model 4 | Model 5 | Model 6 |
|---|---|---|---|---|---|---|
| Covariance (with Science literacy) |  |  |  |  |  | 0.001 |
| BIC | 321309 | 315004 | 231659 | 309186 | 228906 | 226780 |
| N Individuals | 145098 | 145098 | 107366 | 141450 | 105356 | 105356 |
| N Countries | 144 | 144 | 124 | 144 | 124 | 124 |

***p <0.001

**p < .001

*p<0.05

highest and lowest science literacy (left panel), and self-assessed knowledge (right panel) based on the unconditional estimates from models 2 and 4. We report the results from the unconditional model to preserve the full set of countries, as 17 cannot be linked to the country level indicators. Results using the conditional models (3 and 5) are essentially the same and are included in the Supplementary information (S7 Fig in the S1 File).

Fig 2 reveals that those in the highest knowledge groups have more positive attitudes to science. This is true in all countries bar one (Morocco). These results demonstrate that, while the magnitude of the science knowledge-attitude correlation is not uniform across the world, the positive relationship is all but universal, with the one exception perhaps 'proving the rule'.

## Discussion

Our objective in this paper has been to extend what we know about the science knowledge-attitude nexus from the national/regional to the global scale. Previous investigations of this topic have been limited to a relatively small number of predominantly wealthy western contexts. Using a survey covering over 90% of the global population, our results have revealed a near universal positive association between general attitudes to science and both science literacy and self-assessed knowledge, controlling for a range of individual and country level characteristics. It is rare to find such a consistent pattern across countries in comparative surveys, given the multiple sources of error in these complex multinational and multilingual endeavours. It is, of course, possible that the one aberrant case–Morocco–is the result of a data or coding error, although we have been unable to find any evidence that this is the case. It is also noteworthy that the standardised estimate for science literacy in our global analysis is almost identical to that found in the 2008 meta-analysis of Allum et al. [3]. Again, this supports the notion of a culturally invariant, weakly positive relationship between science knowledge and attitudes to science.

A limitation of our study is the science literacy measure. These are usually based on multi-item batteries specifically designed to tap understanding of scientific facts or processes across a range of relevant domains. In contrast, our measure is derived from just three items which were designed with other purposes in mind. That said, we would expect a longer, purpose-built measure to exhibit stronger and more consistent effects compared to our shorter, post-hoc index. Put differently, the limitations of the measure, we contend, make the consistency of the findings across countries all the more surprising. Given the likely presence of random measurement error, we should consider these estimates to represent lower bounds. We are also reassured by the similarity in the correlation with attitudes of the science literacy measure with that of the self-report.

A second limitation of our study is that we are not able to satisfactorily address the question of causality with the data available to us. We have adjusted for a range of plausible

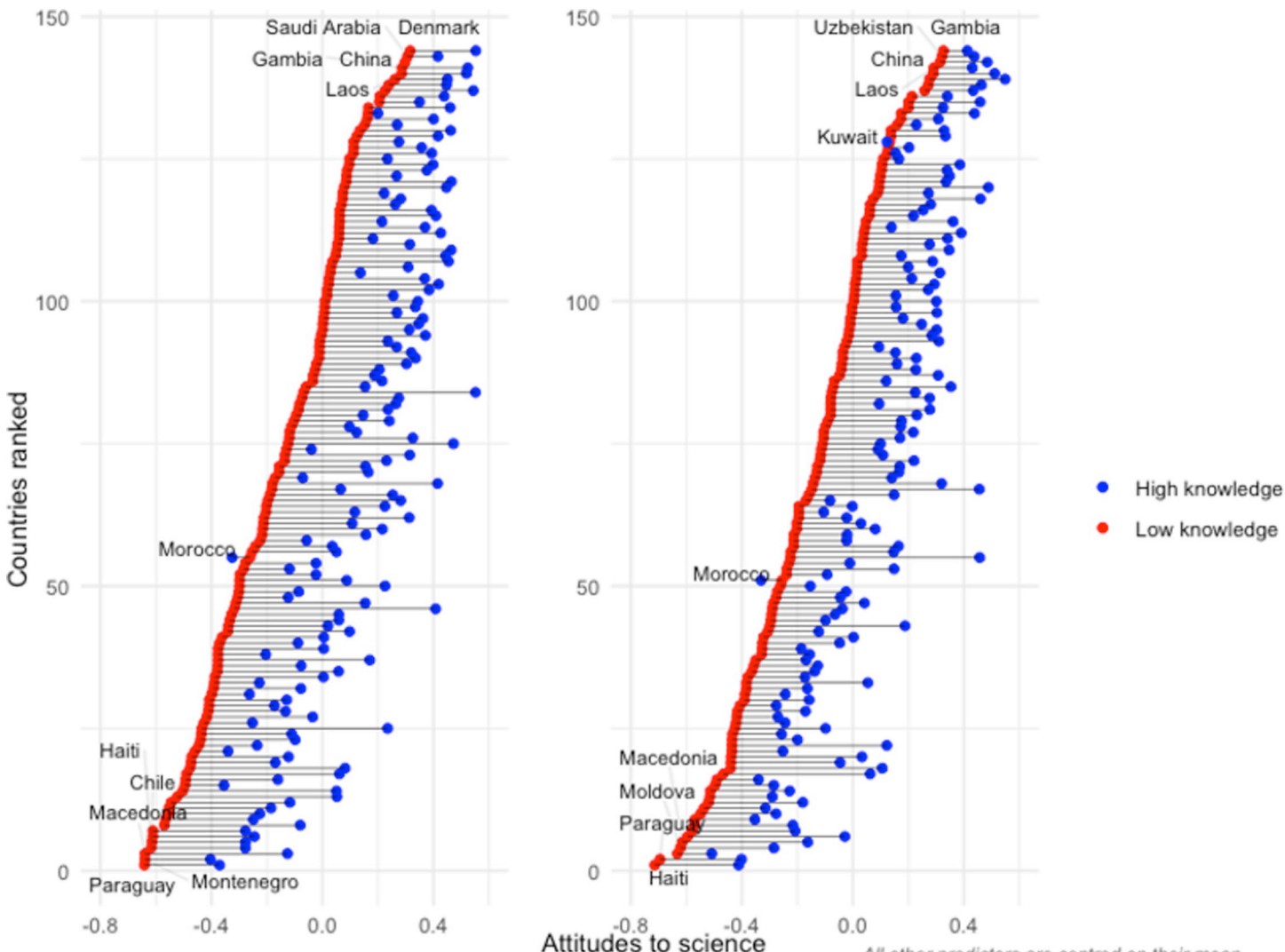

**Fig 2.** Difference in general attitudes to science between high and low science knowledge groups for science literacy (left panel) and self-assessed knowledge (right panel).

confounding variables at both the individual and country levels, but this cannot rule out unobserved variable bias, or the possibility that the causal arrow runs in both directions. It is certainly plausible, for instance, that having a more positive attitude to science motivates people to find out more about it. Nonetheless, the consistency of the relationship across countries attests to its robustness and we hope that these findings will stimulate scholars to better understand the mechanisms underlying the observed regularities in the future.

Science literacy explanations of public attitudes have been criticised for promoting the idea that we might simply educate science scepticism away; an assumption that has received (at best) mixed support in studies of information provision [39]. Moreover, criticisms of the centrality of science literacy have been raised by science communication scholars since the inception of public understanding of science research [40]. Such criticisms point out that knowledge-based explanations of science controversies produce and compound the view that the public is at fault for resistance to scientific research programmes and novel technologies.

This, in turn, motivates paternalism amongst policy makers and delegitimizes often justified criticisms of scientific research programmes.

While such concerns are entirely reasonable, the universality of the knowledge-attitude relationship warrants further investigation of the phenomenon. This study demonstrates that the connection is not merely an artefact of specific characteristics of western societies. Rather, it appears to be a foundational feature of the way publics relate to science across the globe. To be clear, we do not claim that this is the only, or even the most important, way people make sense of science; we acknowledge that such an explanation is intertwined with other roots of science attitudes, such as regulatory arrangements, education systems, media environments, and a broad range of individual differences. A revival of interest in science literacy is therefore justified, not for the sake of disinterring the deficit model but to better understand the causes and consequences of the culturally-invariant science knowledge-attitude correlation.

## Supporting information

**S1 File. Measurement models, descriptive information and conditional models.**
(DOCX)

## Author Contributions

**Conceptualization:** Patrick Sturgis, Ian Brunton-Smith, Nick Allum.

**Methodology:** Patrick Sturgis, Ian Brunton-Smith, Nick Allum.

**Writing – original draft:** Patrick Sturgis, Ian Brunton-Smith, Nick Allum, Simon Fuglsang.

**Writing – review & editing:** Patrick Sturgis, Ian Brunton-Smith, Nick Allum, Simon Fuglsang.

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
