## [Decision Letter · Decision Letter 0]

7 Nov 2023

PONE-D-23-31763Testing the cultural-invariance hypothesis: A global analysis of the relationship between scientific knowledge and attitudes to sciencePLOS ONE

Dear Dr. Sturgis,

Thank you for submitting your this fine and competently done manuscript to PLOS ONE. After careful consideration, we feel that it has merit but does not fully meet PLOS ONE’s publication criteria as it currently stands. Therefore, we invite you to submit a revised version of the manuscript that addresses the points raised during the review process.

Please, refer to reviewers comments where you may see some minor notes on the discussion about potential limitations in using these specific questions to measure the outcomes you are studying.

We look forward to receiving your revised manuscript.

Kind regards,

Paulo Alexandre Azevedo Pereira Santos, PhD

Academic Editor

PLOS ONE

Journal Requirements:

3. We noted in your submission details that a portion of your manuscript may have been presented or published elsewhere. Yes, the Wellcome Monitor survey data is a publicly available data set that can be downloaded from the Wellcome Trust website. Please clarify whether this [conference proceeding or publication] was peer-reviewed and formally published. If this work was previously peer-reviewed and published, in the cover letter please provide the reason that this work does not constitute dual publication and should be included in the current manuscript.

4. We note that Figure 1 in your submission contain map images which may be copyrighted. All PLOS content is published under the Creative Commons Attribution License (CC BY 4.0), which means that the manuscript, images, and Supporting Information files will be freely available online, and any third party is permitted to access, download, copy, distribute, and use these materials in any way, even commercially, with proper attribution. For these reasons, we cannot publish previously copyrighted maps or satellite images created using proprietary data, such as Google software (Google Maps, Street View, and Earth). For more information, see our copyright guidelines: http://journals.plos.org/plosone/s/licenses-and-copyright.

We require you to either present written permission from the copyright holder to publish these figures specifically under the CC BY 4.0 license, or (2) remove the figures from your submission:

Reviewers' comments:

Reviewer's Responses to Questions

**Comments to the Author**

1. Is the manuscript technically sound, and do the data support the conclusions?

Reviewer #1: Yes

Reviewer #2: Yes

2. Has the statistical analysis been performed appropriately and rigorously? 

Reviewer #1: Yes

Reviewer #2: Yes

3. Have the authors made all data underlying the findings in their manuscript fully available?

Reviewer #1: Yes

Reviewer #2: Yes

4. Is the manuscript presented in an intelligible fashion and written in standard English?

Reviewer #1: Yes

Reviewer #2: Yes

5. Review Comments to the Author

Reviewer #1: This WGM survey evaluated 149,000 respondents across 144 countries on the parameter of ‘science literacy’ and respondents’ self-assessments of their understanding of science for better understanding of knowledge-attitude relationship.

The Background and Relevant Literature is reviewed in this paper. The study evaluated how scientific knowledge is related to attitudes to science in global perspective (covering over 90% of the world’s population). The data supported the strength and direction of the knowledge-attitude relationship and its variations across country contexts. I think it`s helpful for readers for better understand the causes and consequences of the culturally-invariant science knowledge-attitude correlation, a nice work!

Reviewer #2: This is a competently done paper, using Multivariate, Multilevel IRT analysis to indicate that, barring one exceptional case, the link between Science Understanding, Subjective Science Understanding, and Trust in Science is positive. The analysis is particularly interesting in that it shows that gaps between trust between low and high "understanders" is strikingly similar across nations covering 90% of the world's population.

I just have a few minor comments. First, although it was discussed in the concluding section, I might add 1-2 sentences noting the limitations of the three questions measuring "objective" scientific knowledge (more on reference to 35-36 given that many might take issue with the limitations of the questions). The authors also may wish to comment a bit more that there is good variation on the questions--I was surprised that public understanding was not higher, particularly on items 1 and 3 of the scale.

On Figure 1--if it comes out in Black and White-you don't get the visual discrimination between countries you see in the colourised version. Looking at the left and right planes of figure 2, I cannot tell hat is different immediately--perhaps more specific labelling?

6. PLOS authors have the option to publish the peer review history of their article (what does this mean?). If published, this will include your full peer review and any attached files.

Reviewer #1: **Yes: **Zakir Uddin

Reviewer #2: **Yes: **Professor Thomas Scotto

---

## [Author Response · Author response to Decision Letter 0]

6 Dec 2023

I just have a few mino comments. First, although it was discussed in the concluding section, I might add 1-2 sentences noting the limitations of the three questions measuring "objective" scientific knowledge (more on reference to 35-36 given that many might take issue with the limitations of the questions). 

Thank you for this suggestion. We have added a new sentence acknowledging the limitation of the questions on p8. 

The authors also may wish to comment a bit more that there is good variation on the questions--I was surprised that public understanding was not higher, particularly on items 1 and 3 of the scale.

We have added a sentence noting this variation on p11. 

On Figure 1--if it comes out in Black and White-you don't get the visual discrimination between countries you see in the colourised version. Looking at the left and right planes of figure 2, I cannot tell hat is different immediately--perhaps more specific labelling?

We agree that this was not clear and have added an explanation of the difference between the 2 charts in the Figure caption.

---

## [Editor Report · Decision Letter 1]

20 Dec 2023

Testing the cultural-invariance hypothesis: A global analysis of the relationship between scientific knowledge and attitudes to science

PONE-D-23-31763R1

Dear Dr. Sturgis,

We’re pleased to inform you that your manuscript has been judged scientifically suitable for publication and will be formally accepted for publication once it meets all outstanding technical requirements.

Kind regards,

Paulo Santos, MD, PhD

Academic Editor

PLOS ONE

---

## [Editor Report · Acceptance letter]

25 Jan 2024

PONE-D-23-31763R1 

PLOS ONE

Dear Dr. Sturgis, 

I'm pleased to inform you that your manuscript has been deemed suitable for publication in PLOS ONE. Congratulations! Your manuscript is now being handed over to our production team.

Kind regards, 

on behalf of

Professor Paulo Alexandre Azevedo Pereira Santos 

Academic Editor

PLOS ONE